# Supramolecular Self-Assembly of Dipalmitoylphosphatidylcholine and Carbon Nanotubes: A Dissipative Particle Dynamics Simulation Study

**DOI:** 10.3390/nano12152653

**Published:** 2022-08-02

**Authors:** Mahboube Keshtkar, Nargess Mehdipour, Hossein Eslami

**Affiliations:** Department of Chemistry, College of Sciences, Persian Gulf University, Boushehr 75168, Iran; m.keshtkar0067@gmail.com (M.K.); heslami@pgu.ac.ir (H.E.)

**Keywords:** carbon nanotubes, dissipative particle dynamics, lipids, self-assembly

## Abstract

Dissipative particle dynamics simulations were performed to investigate the self-assembly of dipalmitoylphosphatidylcholine (DPPC) as a model lipid membrane on the surface of carbon nanotubes (CNTs). The influence of surface curvature of CNTs on self-assembly was investigated by performing simulations on solutions of DPPC in water in contact with CNTs of different diameters: CNT (10, 10), CNT (14, 14), CNT (20, 20), and CNT (34, 34). DPPC solutions with a wide range of concentrations were chosen to allow for formation of lipid structures of various surface densities, ranging from a submonolayer to a well-organized monolayer and a CNT covered with a lipid monolayer immersed in a planar lipid bilayer. Our results are indicative of a sequence of phase-ordering processes for DPPC on the surface of CNTs. At low surface coverages, the majority of hydrocarbon tail groups of DPPC are in contact with the CNT surface. Increasing the surface coverage leads to the formation of hemimicellar aggregates, and at high surface coverages close to the saturation limit, an organized lipid monolayer self-assembles. An examination of the mechanism of self-assembly reveals a two-step mechanism. The first step involves densification of DPPC on the CNT surface. Here, the lipid molecules do not adopt the order of the target phase (lipid monolayer on the CNT surface). In the second step, when the lipid density on the CNT surface is above a threshold value (close to saturation), the lipid molecules reorient themselves to form an organized monolayer around the tube. Here, the DPPC molecules adopt stretched conformations normal to the surface, the end hydrocarbon groups adsorb on the surface, and the head groups occupy the outermost part of the monolayer. The saturation density and the degree of lipid ordering on the CNT surface depend on the surface curvature. The saturation density increases with increased surface curvature, and better-ordered structures are formed on less curved surfaces.

## 1. Introduction

Understanding the self-assembly rules of lipid molecules plays a key role in life sciences. Among the lipid molecules, phospholipids, which are the main components of biological membranes, are of particular importance. In this respect, self-assembly of phospholipids from aqueous solution to form structures ranging in complexity and morphology from spherical vesicles [1] to nanotubes [2] has been widely studied. Such investigations have a diverse range of applications in many areas of science and technology, such as the design of nanomaterials, drug delivery, and controlled release [3].

Inspired by the unique physical properties of carbon nanotubes (NCTs), such as their excellent mechanical strength and chemical stability, light weight, and high thermal conductivity [4,5], researchers have taken advantage of the physical and chemical properties of CNTs in biomedical applications. The hydrophobicity-induced aggregation of CNTs, on the other hand, prevents their applications for such purposes. Over the past two decades, scientists have found that covalent and/or noncovalent surface modification of CNTs prevents their aggregation [6,7]. The noncovalent strategy preserves the inherent properties of nanotubes and is therefore preferable over the covalent alternative. This method employs the hydrophobic interactions between amphiphilic molecules and the CNT surface, as well as the hydrophilic interactions between water and amphiphiles, to dissolve the CNT, i.e., to stabilize the aqueous dispersion. Among possible amphiphiles, phospholipids are regarded as an interesting class of material with respect to formation of supramolecular complexes with CNTs. The resulting supramolecular assemblies take advantage of the physical and chemical properties of the constituent components, i.e., lipids and CNTs. The combination of lipid–lipid, lipid–CNT, and lipid–water interactions guides the self-assembly toward the formation of structures that cannot be obtained in the absence of CNTs. Furthermore surface coating by lipids makes the CNTs biocompatible [8,9].

Experimental investigations have addressed the adsorption of surfactants and lipid molecules dissolved in aqueous solutions on CNTs [10,11,12,13,14]. Transmission electron microscopy experiments have shown that single-tailed phospholipids form supramolecular complexes on the surface of CNTs [10,15]. Recently, self-assembly of phospholipids on CNTs has been shown to result in the formation of double-helical phospholipid-modified nanotube structures [16,17]. Such experimental findings are remarkable in view of the types of structures formed on the surface of CNTs and the potential applications of lipid–CNT hybrids. However, a molecular-level understanding of the adsorption process and the mechanism of structural formation of lipids on the surface of CNTs cannot easily be obtained from experimental results. Computer simulations, on the other hand, allow for a detailed investigation of the mechanism of aggregate formation and the dynamics of self-assembly at the molecular level.

To date, a number of atomistic simulations [18,19,20] have addressed the mechanism of adsorption of single- and double-tailed lipids on CNTs. Owing to the complexity and size of the systems under investigation, coarse-grained simulations [20,21,22] have also been performed to elucidate the mechanism of lipid self-assembly on the surface of CNTs. 

In this work, in order to simulate larger systems over longer time scales than are achievable in atomistic simulations, we employ dissipative particle dynamics (DPD) simulations. DPD is a coarse-grained method, which, upon careful parametrization, can provide meaningful results. Because phosphatidylcholines are the most important component of living organisms, we perform our simulations on a model membrane, dipalmitoylphosphatidylcholine (DPPC). DPPC is known as a useful model for understanding the physical properties of biological membranes [23,24]. We examine the effects of lipid concentration and surface curvature of CNTs on the structural evolution of lipid aggregates at the CNT interface and investigate the mechanism of self-assembly.

## 2. Model and Simulation Details

We employ a coarse-grained (DPD) model to describe lipid molecules, the solvent (water), and CNTs. DPD is a particle-based simulation scheme in which the motion of particles is governed by Newton’s law. There are three components of force considered in this method [25]: a pairwise, nonbonded, conservative force; a random force; and a dissipative force. The conservative force between two DPD beads, *i* and *j*, reads as:(1)$$F_{ij}^{C}=a_{ij}\left(1-\frac{r_{ij}}{r_{C}}\right){\hat{r}}_{ij}\;$$
where $r_{C}$ is the cutoff distance, $r_{ij}$ is the distance between beads *i* and *j*, ${\hat{r}}_{ij}=r_{ij}/\;r_{ij}$ is a unit vector, and the $a_{ij}$ is the maximum repulsion. The expressions for random and dissipative forces are as follow:(2)$$F_{ij}^{R}=\sigma \left(1-\frac{r_{ij}}{r_{C}}\right)\theta_{ij}∆t^{-1/2}{\hat{r}}_{ij}\;$$
and
(3)$$F_{ij}^{D}=-\gamma {\left(1-\frac{r_{ij}}{r_{C}}\right)}^{2}\left({\hat{r}}_{ij}\cdot v_{ij}\right){\hat{r}}_{ij}\;$$
where σ is the noise strength, $\theta_{ij}$ is a random variable with a Gaussian distribution and unit variance, ∆t is the time step, γ is the dissipation strength, and $v_{ij}\;$ is the relative velocity vector between *i* and *j*. Whereas random force heats up the system, the dissipative force cools it down. In other words, the combination of random and dissipative forces acts as the DPD thermostat. According to the fluctuation–dissipation theorem, noise and dissipation strengths are linked by the relation $\sigma^{2}=2\gamma k_{B}T$ ($k_{B}T$ being the thermal energy per bead) [25]. Owing to its computational efficiency, the DPD method has been widely employed for simulation of fluids of a variety of molecular shapes and complexities [26,27].

In our model, a combination of four water molecules is regarded a single DPD bead. The volume of four water molecules corresponds to 0.12 nm^3^. We performed our DPD simulations at a reduced density of ρ=3. This means that there are three water molecules in a cube of volume ($r_{C}^{3}$); therefore, in our simulations, $r_{C}=0.71\;\mathrm{nm}$. In this work, the parameters $r_{C}$ and $k_{B}T$ are used to reduce the length and energy parameters, respectively.

The DPD model for DPPC is shown in Figure 1. Each lipid molecule consists of 12 DPD beads (4 head-group and 12 tail-group beads). The DPPC beads are connected by harmonic bonds:(4)$$u_{\mathrm{bond}}=\frac{1}{2}k_{\mathrm{bond}}{\left(r-r_{ij}^{0}\right)}^{2}$$
where $u_{\mathrm{bond}}$ is the bond potential, $k_{\mathrm{bond}}$ is the force constant, and $r_{ij}^{0}$ is the equilibrium bond length for the bond between beads *i* and *j*. The angle potential, $u_{\mathrm{angle}}$, is expressed as:(5)$$u_{\mathrm{angle}}=\frac{1}{2}k_{\mathrm{angle}}\left(1-\cos\left(\theta_{ijk}-\theta_{ijk}^{0}\right)\right)$$
where $k_{\mathrm{angle}}$ is the force constant, and $\theta_{ijk}^{0}$ is the equilibrium angle.

The DPD interaction parameters for DPPC were recently parameterized in Ref. [24]. The mapping scheme for CNT is based on a previously developed coarse-grained model of graphene [28,29], in which eight C atoms are regarded as a single bead.

The DPD repulsion parameter for water in the present four-to-one mapping, $a_{\mathrm{ww}}=100$, reproduces the compressibility of water at room temperature. Therefore, all DPD repulsion parameters for beads of the same type were set to 100, and the cross-interaction parameters were determined according to the Flory−Huggins χ-parameter. In order to avoid dealing with explicit electrostatic interactions in DPD [30], we slightly increased the repulsion strength between the similarly charged head groups of DPPC ($a_{h1h1}=a_{h2h2}$ = 110) to compensate for electrostatic repulsion [24]. The DPPC bilayer model was previously validated [24] by simulating a DPPC bilayer immersed in pure water and comparing the area per lipid molecule and the bilayer thickness against experiment and existing atomistic simulation results in the literature [23]. The DPD repulsion parameters ($a_{ij}$), as well as the parameters for bonded potentials, are reported in Table 1.

We simulated 44 systems, the details of which are tabulated in Table 2. Initial solutions of DPPC in water were placed in contact with CNTs of varying diameters. The number of DPPC molecules ranged from 10 to 500. In all samples, the number of water beads was 20,000. In order to study the effect of surface curvature on self-assembly, we simulated DPPC–water systems containing CNT (10, 10), CNT (14, 14), CNT (20, 20), and CNT (34, 34) with diameters of ≈ 1.97, ≈2.67, ≈3.94, and ≈6.2, respectively. The tube length in all simulations was ≈ 15.5, and the DPD tube beads were fixed in position during the simulations. The tube was aligned along the *y* axis, and the box size along the *y* axis (*L_y_*) was kept close to 22.5 (larger than the tube length) in all simulations. All simulations were performed in an isothermal–isobaric ensemble. The simulation box size in the *x* and *z* directions (perpendicular to the tube axis) was allowed to change independently to keep the perpendicular components of pressure fixed. All simulations were performed at 323 K above the gel-to-liquid crystalline temperature of the DPPC model membrane (315 K) [24] using the YASP simulation package [31]. All the starting configurations were established by randomly inserting lipid molecules and water in a large simulation box containing a single CNT, with no overlap between lipid, water, and CNT beads. No water or lipid beads were allowed to be inserted inside the CNT. A wide range of concentrations of DPPC in water were chosen (see Table 2). We performed long DPD simulations (1.3 × 10^5^, corresponding to 500 ns) to achieve equilibration. Equilibration was monitored as the stage where the number of DPPC molecules in the solution becomes constant (close to zero).

## 3. Results and Discussion

### 3.1. Clustering and Surface Adsorption of DDPC

With the shrinkage of box size during the course of simulation, the DPPC molecules in the solution form dimers, trimers, and larger aggregates. Aggregate formation is interpreted in terms of higher concentrations of DPPC in solution in our samples than the critical micelle concentration of DPPC (0.5 nM) [20]. A snapshot of simulation box showing aggregate formation in DPPC is shown in Figure 2.

Owing to the favorable hydrophobic interactions between the tail groups of DPPC and the CNT, as well as the hydrophilic head group-water interactions, the DPPC hydrocarbon tails adsorb on the surface of the CNT. Both monomers and larger aggregates adsorb on the CNT surface, and the lipid molecules reorient on the CNT surface to form stable configurations, depending on the DPPC concentration and the CNT diameter. We have shown the time-dependence of the fraction of initial DPPC molecules in the solution, which are adsorbed on the surface of CNT (10, 10), CNT (14, 14), CNT (20, 20), and CNT (34, 34), as shown in Figure 3. A lipid molecule is defined as adsorbed on the surface of the CNT if at least one of its constituent beads is within a distance of 1.06 from the surface. The monotonic increase in the fraction of surface-adsorbed lipid molecules with time indicates one-by-one addition of lipids from the solution to the CNT surface. However, the jump in the fraction of surface-adsorbed lipids indicates joining of the large clusters to the surface. As larger-diameter CNTs provide a larger surface area, the rate of surface adsorption is higher on the surface of larger-diameter CNTs. Fluctuations in the fraction of adsorbed lipid molecules in equilibrium reveals that some of the surface-adsorbed molecules are exchanged between the solution and the CNT surface. Expectedly, larger fluctuations are observed on the surface of more curved CNTs, which provide less attraction for lipid adsorption than less curved surfaces [32,33]. 

### 3.2. Surface Saturation of CNTs with DPPC

The number of lipid molecules in the solution (see Table 2) varies over a wide range. At the low-concentration limit, only a submonolayer lipid forms on the CNT surface. However, at the highest concentrations investigated in this work, a complete monolayer forms around the CNT (the CNT surface saturates with DPPC), and some excess lipid molecules form a bilayer in contact with the surface-coated CNT in the solution. Figure 4 shows snapshots of a simulation box, revealing CNT surface saturation with DPPC and formation of a planar bilayer in the solution. This indicates that at concentrations higher than the saturation limit, formation of a planar bilayer in the system is more favorable than formation of a second layer (bilayer) around the tube. This is in agreement with simulation results of the penetration of CNTs in DPPC [34]; parallel (to the bilayer) insertion of the CNT does not affect the bilayer stability, as the bilayer is able to self-seal. In the systems containing CNT (34, 34), owing to the large diameter of tube, the lipid molecules also adsorb on the inner surface (inside) of the tube. 

Based on the examined DPPC concentrations and the types of CNTs in the solution, we calculated the amount of DPPC needed to saturate the surface of the CNT. The results in Figure 5 show that at saturated coverage, the surface density increases with decreased tube diameter. The head groups of DDPC are larger than the tail groups. There is more space on the outer surface of more curved (smaller diameter) tubes to accommodate larger head groups, which occupy the outermost part of the monolayer. This explains the increase in the surface density of DPPC with increased surface curvature of the CNT.

### 3.3. Ordering of DPPC Molecules on the CNT Surface

At concentrations lower than the saturated coverage conditions, the CNT surface is not fully covered by DPPC molecules. At low surface coverages, the favorable tail-group–CNT attractions cause parallel (to the CNT surface) orientations of DPPC molecules. In such conformations, the hydrophobic tail groups of DPPC attach to the CNT surface. and its hydrophilic head groups orient away from the surface (see Figure 6a) to be hydrated by the surrounding water molecules. Increasing the DPPC concentration in the solution increases the CNT surface coverage. The adsorbed DPPC molecules on the surface of the CNT adopt conformations in which the lipid tail groups are in contact with the CNT, whereas the head groups are in contact with each other and with the aqueous solution. This structure is similar to that of hemimicelles in solution [30]. In the hemimicellar structures on the CNT surface, an almost random orientation of lipid molecules is observed (Figure 6b). However, at densities close to saturation limit, the DPPC molecules adopt ordered conformations normal to the CNT surface (see Figure 6c,d). This reveals that the self-assembly of lipid molecules on the surface of CNTs involves a sequence of processes. At low surface coverages, the surface-adsorbed lipid molecules adopt extended structures on the surface; increasing the surface coverage leads to the formation of hemimicellar aggregates; finally, at high surface coverages, an organized lipid monolayer self assembles. To reveal the nature of this transition in detail, in the following, we quantified the order as a function of surface density. 

Figure 7 shows the (number) density profile of DPPC molecules adsorbed on the surface of CNT (34, 34) as a function of distance from the surface. We discuss the results in terms of the saturation density ($\rho_{s}=$1.074). At low surface densities (≈0.5$\rho_{s}$), the lipid molecules adsorbed on the CNT surface are found in close vicinity of the surface. With increased surface density, some DPPC molecules stretch away from the CNT surface. However, the effect becomes more pronounced at densities close to the saturation (>0.9$\rho_{s}$).

Our findings show that the self-assembly of lipid molecules into an organized monolayer structure, such as the structure shown in Figure 6d, only occurs at high surface coverages (close to saturation). The first stage of self-assembly involves accumulation/aggregation of lipid molecules on the CNT surface. However in this stage, the lipid molecules do not adopt an orientation (normal to the CNT surface), in accordance with an organized monolayer. Only at high surface densities does the packing entropy cause lipid chains to stretch away from the CNT surface, i.e., to self-assemble into an organized monolayer. 

### 3.4. Mechanism of Lipid Ordering on the CNT Surface

Figure 8 shows the quantified order of lipid molecules adsorbed on the surface of CNT (20, 20) immersed in aqueous solutions containing 100, 200, 300, 400, and 500 DPPC molecules. The order is quantified in terms of the angle, θ, between two unit vectors: a unit vector along the hydrocarbon tail groups of each chain and a unit vector normal to the CNT surface. Here, $\theta =0^{{}^{\circ}}$ corresponds to the perpendicular alignment of DPPC molecules around the tube, and $\theta =90^{{}^{\circ}}$ is tangent to the CNT surface orientations. The order for two hydrocarbon chains of each lipid molecule is averaged. The normalized probability distribution functions for the orientation of DPPC molecules adsorbed on the CNT surface (see Figure 8) show that at low surface coverages (≈0.5$\rho_{s}$), a fraction of lipid molecules adopt tilted conformations around the tube. Increasing the surface density improves the lipid ordering around the tube, although a noticeable order is observed at high surface densities, i.e., above 0.9$\rho_{s}$ (see Figure 8 at ρ≥ 1.19). With a further increase in the surface density, highly ordered lipid chains form around the tube. 

This is reminiscent of a two-step mechanism of nucleation/solidification previously described in phase ordering of colloidal particles [35]. The first step of this two-step mechanism involves densification. In this step, the lipid molecules occupy the CNT surface, but they do not adopt the order of the target phase (lipid monolayer on the CNT surface). In other words, during this step, the positional ordering (densification) occurs prior to orientational ordering. Entropically, it is more favorable for the low-density lipid chains to adopt random orientations on the CNT surface. When the density increases to a threshold value close to that of the target phase, the lipid molecules reorient themselves to form an ordered structure on the CNT surface. In this second step, the density does not increase noticeably; rather, the change in orientational ordering is the dominant process. This phase ordering at high densities increases the packing entropy [35,36] of lipid chains.

## 4. Conclusions

We performed dissipative particle dynamics (DPD) simulations to investigate the self-assembly of dipalmitoylphosphatidylcholine (DPPC) as a model lipid membrane on the surface of carbon nanotubes (CNTs). The DPD interaction parameters were well-tuned against bilayer thickness and surface area per lipid molecule for a DPPC bilayer immersed in pure water [24]. We simulated a number of systems, each with a single CNT immersed in solutions of DPPC in water. The concentration of DPPC in water was varied over a wide range to allow for a wide range of CNT surface coverages: from a submonolayer to a well-organized monolayer and a lipid-covered CNT immersed in a planar lipid bilayer. This allowed us to study the mechanism of the phase-ordering process, depending on the surface density of the lipid. Moreover, in order to study the effect of surface curvature on self-assembly, we simulated systems containing CNTs of varying diameters: CNT (10, 10), CNT (14, 14), CNT (20, 20), and CNT (34, 34). 

The DPPC molecules in the solution form dimers, trimers, and larger aggregates. The favorable hydrophobic interactions between the hydrocarbon tails of DPPC and CNT, as well as hydrophilic head-group–water interactions lead to surface adsorption of DPPC. In the early stages of self-assembly, aggregates of all size may adsorb on the CNT surface. The adsorbed lipid molecules then adopt an equilibrium surface packing/orientation, depending on the degree of surface coverage. At the low concentration (of DPPC in solution) limit, only a submonolayer lipid forms on the CNT surface. However, at high concentrations, the CNT surface is fully covered with an organized DPPC monolayer. This saturation surface density depends on the CNT diameter (surface curvature); it increases with increased surface curvature. This is interpreted in terms of the larger size of DPPC head groups compared to tail groups. A larger space is accessible to larger head groups on more curved surfaces. With a further increase in the initial concentration of DPPC in solution (above the saturation density of the CNT surface), the excess lipid molecules form a bilayer in contact with the surface-coated CNT in the solution. This finding is in agreement with previous simulation results of the penetration of CNTs in DPPC [34], where the bilayer remains stable upon parallel (to the bilayer) insertion of a CNT. 

Depending on the initial concentration of DPPC in the solution (surface coverage), a sequence of phase-ordering processes occurs on the CNT surface. At low surface coverages, the favorable tail-group–CNT attractions cause parallel (to the CNT surface) orientations of DPPC molecules. The majority of hydrophobic tail groups of DPPC attach to the CNT surface, and the hydrophilic head groups orient away from the surface. With increased surface coverage, DPPC hemimicellar structures with an almost random orientation of lipid molecules with respect to the CNT surface form on the surface. However, at densities close to the saturation limit, the DPPC molecules form an organized monolayer around the tube. Here, the DPPC molecules adopt stretched conformations normal to the surface, the end hydrocarbon tails adsorb on the CNT surface, and the head groups occupy the outermost part of the monolayer. 

Our findings are indicative of a two-step mechanism of self-assembly of lipids on the CNT surface. In the first step, dense DPPC aggregates form on the CNT surface. In this stage, the adsorbed lipid molecules do not adopt the order of the target phase (lipid monolayer on the CNT surface). In the second step, wherein the lipid density on the CNT surface is above a threshold value (close to saturation), the lipid molecules reorient themselves to form an ordered monolayer around the tube. In this step, the density does not increase noticeably; rather, the change in orientational ordering is the dominant process. This phase ordering at high densities increases the packing entropy [35,36] of lipid chains.

## Figures and Tables

**Figure 1 nanomaterials-12-02653-f001:**
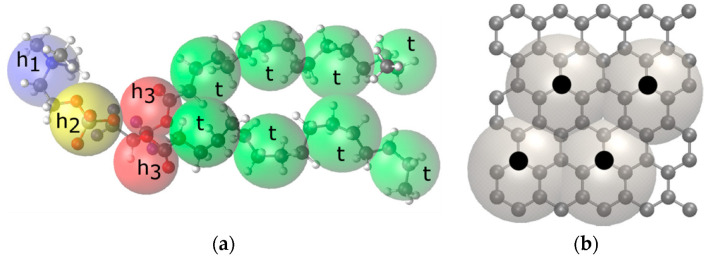
The mapping scheme adopted in this work for (**a**) dipalmitoylphosphatidylcholine (DPPC) and (**b**) carbon nanotubes. Each DPPC molecule is lumped into 12 beads: four head groups (h_1_, h_2_, and h_3_) and eight tail groups (t). The black points in panel (**b**) represent the centers of beads.

**Figure 2 nanomaterials-12-02653-f002:**
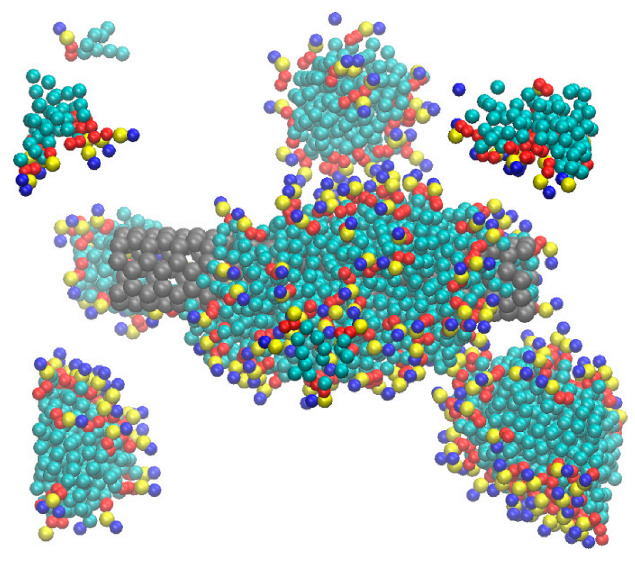
Snapshot of the simulation box, indicating clustering of DPPC molecules in the solution. Both free DPPC molecules (monomers) and clusters adsorb on the surface of CNT (14, 14). Water molecules are not shown.

**Figure 3 nanomaterials-12-02653-f003:**
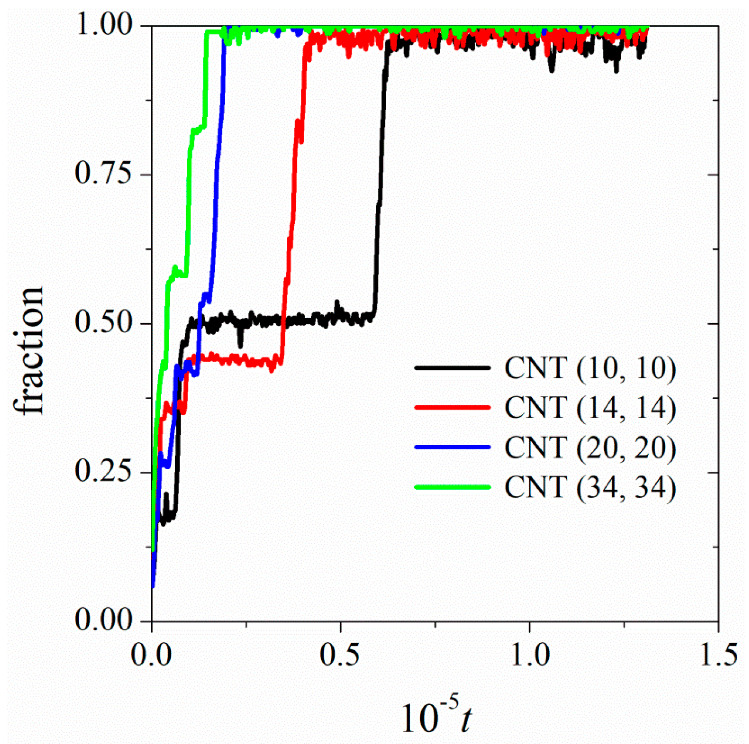
Time dependence of the fraction of DPPC molecules adsorbed on the surface of CNTs. All systems contain 250 lipid molecules, and the type of CNT is shown in the figure legend.

**Figure 4 nanomaterials-12-02653-f004:**
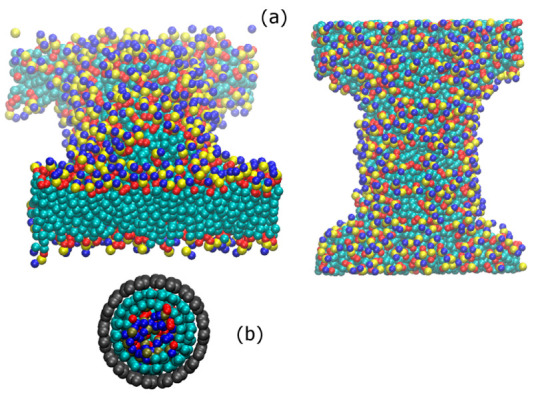
Snapshots of a simulation box showing (**a**) surface saturation of CNT (20, 20) with DPPC molecules and formation of a bilayer, as well as (**b**) adsorption of lipid molecules on the inner surface of CNT (34,34). In panel (**a**), the left and right snapshots indicate views along the tube (*y*) and normal to the tube (*z*) axes, respectively. The DPPC molecules dissolved in the solution (500 molecules) adsorb on the surface of the CNT to form a complete monolayer. The excess DPPC molecules form a bilayer at the CNT ends, spanning the simulation box in the *x* and *y* directions.

**Figure 5 nanomaterials-12-02653-f005:**
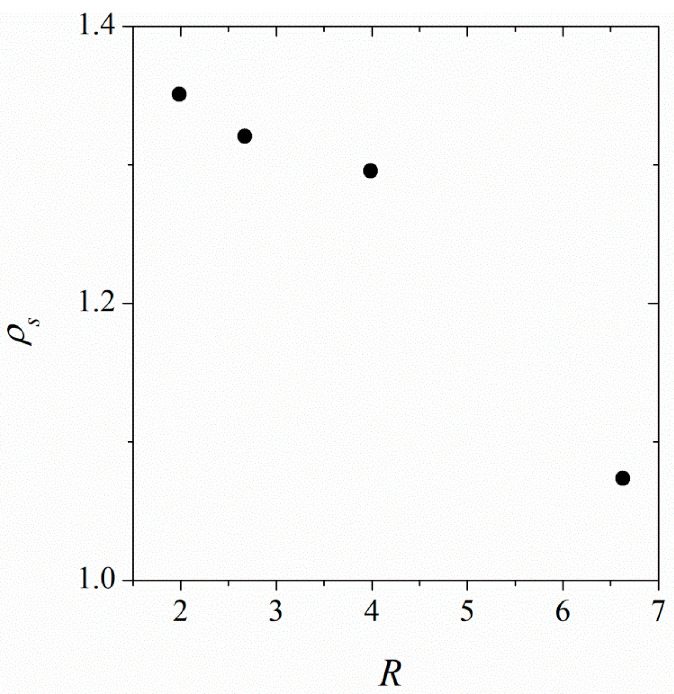
The surface density of DPPC at saturation, $\rho_{s}$, as a function of CNT diameter, *R*.

**Figure 6 nanomaterials-12-02653-f006:**
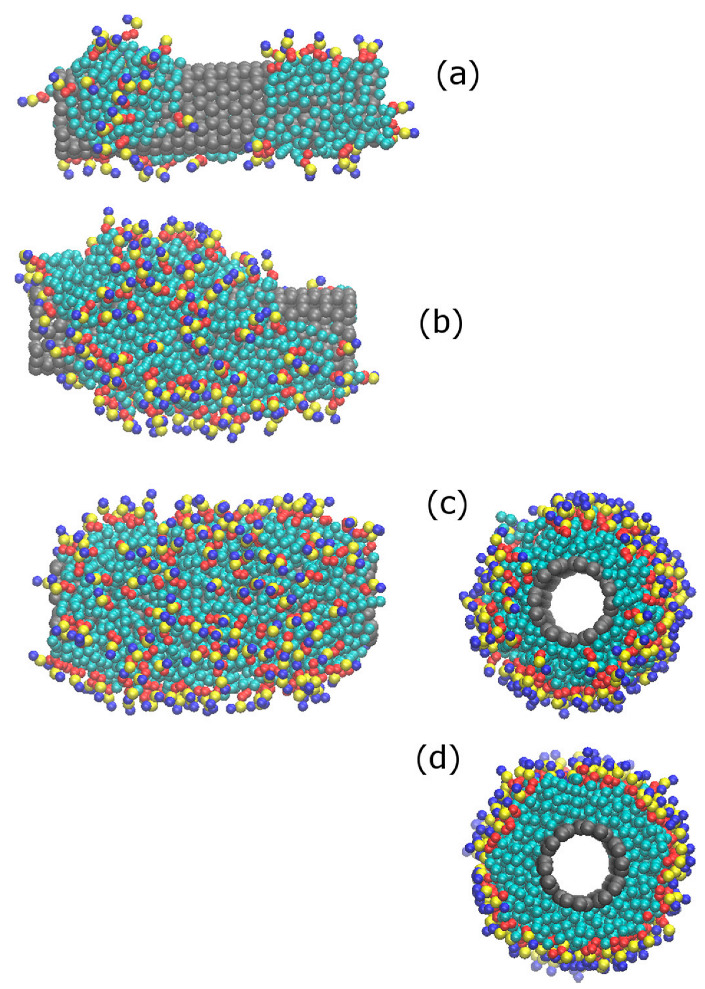
Snapshots of a simulation box indicating surface density dependence of ordering of DPPC lipid molecules around the CNT (20, 20). In snapshots (**a**–**d**), the surface densities of lipid molecules are 0.28, 0.83, 1.05, and 1.22 (corresponding to 0.56 nm^−2^, 1.64 nm^−2^, 2.09 nm^−2^, and 2.43 nm^−2^, respectively).

**Figure 7 nanomaterials-12-02653-f007:**
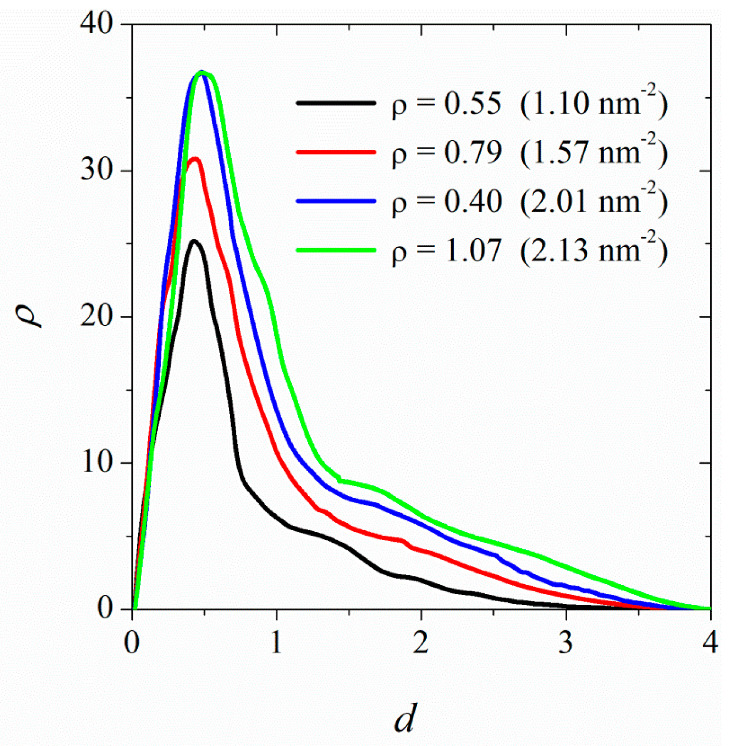
Number density profiles for DPPC molecules adsorbed on the surface of CNT (34, 34) as a function of distance from the CNT surface. The surface densities are shown in the figure legend. The highest density corresponds to surface saturation.

**Figure 8 nanomaterials-12-02653-f008:**
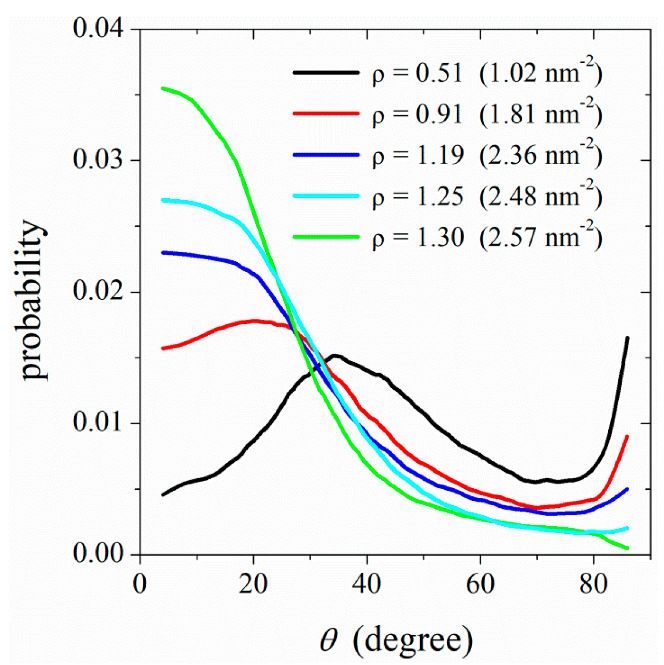
Surface density dependence of the orientation of lipid molecules adsorbed on the surface of CNT (20, 20). The CNT is immersed in solutions initially containing 100, 200, 300, 400, and 500 lipid molecules. The orientation angle is defined as the angle between a unit vector parallel to the tail-group beads and a unit vector normal to the CNT surface.

**Table 1 nanomaterials-12-02653-t001:** DPD parameters for DPPC, water, and carbon nanotubes (CNTs).

(**a**) Bonded potential						
bond type *i*-*j*		$k_{ij}$ ($k_{B}T/r_{C}^{2}$)		$r_{ij}^{0}$ ($r_{C}$)		
h_1_–h_2_		512		0.47		
h_2_–h_3_		512		0.47		
h_3_–h_3_		512		0.31		
h_3_–t		512		0.59		
t–t		512		0.59		
(**b**) Angle-bending potential						
angle type *i*-*j*-*k*		$k_{k_{\mathrm{angle}}}$ ($k_{B}T$)		$\theta_{ijk}^{0}$(degree)		
h_2_–h_3_–h_3_		6		120.0		
h_2_–h_3_–t		6		180.0		
h_3_–t–t		6		180.0		
t–t–t		6		180.0		
(**c**) Repulsion parameters, $a_{ij}$, for all bead types						
	h_1_	h_2_	h_3_	t	w	CNT
h_1_	110	100	102	130	98	130
h_2_	100	110	102	130	98	130
h_3_	102	102	100	110	102	110
t	130	130	110	100	130	100
w	98	98	102	130	100	130
CNT	130	130	110	100	130	100

**Table 2 nanomaterials-12-02653-t002:** Details of the systems simulated in this work.

Substrate	Number of DPPC Lipid Molecules	Equilibrium Box Size
CNT (10, 10)	10	17.56 × 22.53 × 17.39
	50	17.66 × 22.55 × 17.68
	100	17.89 × 22.52 × 17.93
	150	18.20 × 22.53 × 18.10
	200	18.38 × 22.53 × 18.39
	250	18.66 × 22.55 × 18.56
	300	18.95 × 22.55 × 18.73
	350	18.51 × 22.56 × 19.62
	400	18.99 × 22.41 × 19.72
	450	18.91 × 22.48 × 20.18
	500	19.03 × 22.56 × 20.44
CNT (14, 14)	10	17.49 × 22.55 × 17.52
	50	17.72 × 22.55 × 17.69
	100	17.91 × 22.53 × 17.97
	150	18.17 × 22.53 × 18.18
	200	18.41 × 22.52 × 18.42
	250	18.65 × 22.55 × 18.66
	300	18.93 × 22.45 × 18.88
	350	19.08 × 22.58 × 19.07
	400	19.27 × 22.65 × 19.28
	450	19.62 × 22.49 × 19.51
	500	19.70 × 22.56 × 19.79
CNT (20, 20)	10	17.73 × 22.53 × 17.41
	50	17.65 × 22.55 × 17.86
	100	17.96 × 22.55 × 18.03
	150	18.32 × 22.59 × 18.14
	200	18.41 × 22.59 × 18.51
	250	18.70 × 22.55 × 18.66
	300	18.92 × 22.45 × 18.99
	350	19.08 × 22.59 × 19.15
	400	19.31 × 22.66 × 19.31
	450	18.44 × 22.51 × 20.83
	500	18.37 × 22.58 × 21.31
CNT (34, 34)	10	17.76 × 22.55 × 17.56
	50	18.15 × 22.55 × 17.56
	100	18.19 × 22.53 × 17.99
	150	18.32 × 22.53 × 18.32
	200	18.58 × 22.52 × 18.52
	250	18.66 × 22.53 × 18.90
	300	18.97 × 22.55 × 19.08
	350	18.90 × 22.59 × 19.49
	400	19.66 × 22.44 × 19.31
	450	19.41 × 22.52 × 19.91
	500	19.80 × 22.59 × 19.89

## Data Availability

The data presented in this study are available on request from the corresponding author.

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
