# Peer review of "Supramolecular Self-Assembly of Dipalmitoylphosphatidylcholine and Carbon Nanotubes: A Dissipative Particle Dynamics Simulation Study"

_nanomaterials, 2022, doi:10.3390/nano12152653_

Round 1

Reviewer 1 Report

This is a good study; it will help to understand better the mechanism interaction of DPPC with the carbon nanotubes. I understand the research wanted to study different CNTs diameters: CNT (10, 10), CNT (14, 14), CNT (20, 20), and CNT (34, 34). However, in the lab practice, it is more common to work with the chiral CNTs in between zigzag (0,17) to armchair (10, 10), but the diameter is not affected that much. However, this will be a completely new study and paper. 

I have a few minor comments:

The figure 5. The plot is ? vs R, however R is close to 50 nm, how it is possible if you do not have any CNTs with this size? 

This comment is regarding to the previous comment, in the paragraph line 279-184, it says because of the big diameter the lipids are absorbed inside the CNTs. is this for the 50 nm? and can you give me a value of the DPPC length?

Finally, I would like to see a cross-section (snapshot) for the DPPC absorbed inside the CNTs.

The authors did a great job, and it should be published, but some minor corrections are needed.

Author Response

First of all we thank the referee for his/her helpful comments. Here are the reply to the referee's comments and the revisions made in the manuscript according to the comments.

Comment 1: The figure 5. The plot is ? vs R, however R is close to 50 nm, how it is possible if you do not have any CNTs with this size? 

Answer: There was a mistake in the dimension of R in the original version of Figure 5 (instead of Ångstrom, it was written nm). Therefore, the maximum R is 5 nm. However, as the referee has pointed out, in experiment one does not normally deal with nanotubes of such a large diameter. Our results, however, are useful for the investigation of the effect of surface curvature on the adsorption of lipid molecules on the surface. Meanwhile, we point out that an infinitely large-diameter tube corresponds to a flat surface (in this case, graphene). In other words, the results for the tubes of large diameters can to a good extent show the trend of lipid adsorption on the surface of graphene.    

Comment 2: This comment is regarding to the previous comment, in the paragraph line 279-184, it says because of the big diameter the lipids are absorbed inside the CNTs. is this for the 50 nm? and can you give me a value of the DPPC length?

Answer: The thickness of a DPPC bilayer (our model) is 4.7 nm; i.e., the length of the DPPC in organized monolayers corresponds to about 2.35 nm. Therefore, adsorption of DPPC on the inner surface of CNT(34,34) is reasonable.  

Comment 3: Finally, I would like to see a cross-section (snapshot) for the DPPC absorbed inside the CNTs.

Reviewer 2 Report

The article «Supramolecular Self-Assembly of Dipalmitoylphosphatidylcholine and Carbon Nanotubes: A Dissipative Particle Dynamics Simulation Study» is devoted to DPD modelling of adsorption of Dipalmitoylphosphatidylcholine  (DPPC) molecules on the surface of carbon nanotubes (CNTs) with different diameters and their self-organization. The paper is well written, the model and technique of the experiment are justified, the conclusions are interesting. The paper deserves to be published in Nanomaterials, but still there are comments that need to be taken into account.

1.     The scheme of CNTs  mapping should be outlined, albeit briefly, and the parameter by which they are characterized should be described. Just references to literary sources are definitely not enough;

2.     In computer modelling  papers  (molecular dynamics, DPD), it is customary to describe all values in internal units, say, the length in the size of beads or cut-off distance. In this regard, all dimensions, which are equilibrium box  size (Tab.2), CNT diameter R should be reassigned (not nm, but bead size).  

3.     The potential for the bond length sets the corresponding force and is part of the force field. It should be described in the text of the article as, say, equation (4). The same for the angle bending  potential.

4.     Figure 3. Here are presented data for CNTs with different sizes and for the same number of DPPC molecules in the box. I believe that the difference in the behavior, say, significant fluctuations  for (10,10) CNTs,  is caused by differences in surface density.  I believe that it is desirable here to compare the adsorption curves for cases with equal relative, per unit area for given CNTs, numbers of lipid molecules. 

Author Response

First of all we thank the referee for his/her helpful comments. Here are the reply to the referee's comments and the revisions made in the manuscript according to the comments.

Referee #2

The article «Supramolecular Self-Assembly of Dipalmitoylphosphatidylcholine and Carbon Nanotubes: A Dissipative Particle Dynamics Simulation Study» is devoted to DPD modelling of adsorption of Dipalmitoylphosphatidylcholine  (DPPC) molecules on the surface of carbon nanotubes (CNTs) with different diameters and their self-organization. The paper is well written, the model and technique of the experiment are justified, the conclusions are interesting. The paper deserves to be published in Nanomaterials, but still there are comments that need to be taken into account.

  1. The scheme of CNTs  mapping should be outlined, albeit briefly, and the parameter by which they are characterized should be described. Just references to literary sources are definitely not enough;

Answer: We have shown the mapping scheme of CNT in Figure 1.

  1. In computer modelling  papers  (molecular dynamics, DPD), it is customary to describe all values in internal units, say, the length in the size of beads or cut-off distance. In this regard, all dimensions, which are equilibrium box  size (Tab.2), CNT diameter R should be reassigned (not nm, but bead size).  

Answer: All the units in the revised version of the manuscript are reduced. We have written a sentence on page 3 (before Eq. (4)) to show this. Figures 3, 5, 7, and 8 as well as Table 2 were updated accordingly. All the units in the text were changed according to the reviewer’s comments.   

  1. The potential for the bond length sets the corresponding force and is part of the force field. It should be described in the text of the article as, say, equation (4). The same for the angle bending  potential.

Answer: Expressions for the bond and angle potentials were included in the text (page 3).  

  1. Figure 3. Here are presented data for CNTs with different sizes and for the same number of DPPC molecules in the box. I believe that the difference in the behavior, say, significant fluctuations  for (10,10) CNTs,  is caused by differences in surface density.  I believe that it is desirable here to compare the adsorption curves for cases with equal relative, per unit area for given CNTs, numbers of lipid molecules. 

Answer: As for the larger diameter tubes, the DPPC also adsorbs on the inner surface of the tube, it is not possible to provide results at the same DPPC concentration per unit area for all the tubes. We have written in the text (page 7) that larger fluctuations are seen at the surface of more curved CNTs, which provide less attraction for the lipid adsorption than less curved surfaces.